# Enhancing T Cell and Antibody Response in Mucin-1 Transgenic Mice through Co-Delivery of Tumor-Associated Mucin-1 Antigen and TLR Agonists in C3-Liposomes

**DOI:** 10.3390/pharmaceutics15122774

**Published:** 2023-12-14

**Authors:** Ameneh Arabi, Shahab Aria (Soltani), Brandon Maniaci, Kristine Mann, Holly Martinson, Max Kullberg

**Affiliations:** 1WWAMI School of Medical Education, University of Alaska Anchorage, 3211 Providence, Anchorage, AK 99508, USA; aarabi@alaska.edu (A.A.); ssoltani@hivresearch.org (S.A.); blmaniaci@alaska.edu (B.M.); hamartinson@alaska.edu (H.M.); 2Johns Hopkins Medicine, Johns Hopkins University, 1551 Jefferson St., Baltimore, MD 21287, USA; 3Henry M. Jackson Foundation for the Advancement of Military Medicine, 503 Robert Grant Ave, Silver Spring, MD 20910, USA; 4Department of Immunology, Mayo Clinic, 200 First St. SW, Rochester, MN 55905, USA; 5Department of Biological Sciences, University of Alaska Anchorage, Anchorage, AK 99508, USA; kmann1@alaska.edu

**Keywords:** antigen presenting cell, vaccine, cancer immunotherapy, complement C3, liposome, nanoparticle, targeted delivery, toll-like receptor agonist, Mucin-1

## Abstract

Mucin-1 (MUC1) is a highly relevant antigen for cancer vaccination due to its overexpression and hypo-glycosylation in a high percentage of carcinomas. To enhance the immune response to MUC1, our group has developed C3-liposomes that encapsulate the MUC1 antigen along with immunostimulatory compounds for direct delivery to antigen-presenting cells (APCs). C3-liposomes bind complement C3, which interacts with C3-receptors on APCs, resulting in liposomal uptake and the delivery of tumor antigens to APCs in a manner that mimics pathogenic uptake. In this study, MUC1 and Toll-like receptor (TLR) agonists were encapsulated in C3-liposomes to provoke an immune response in transgenic mice tolerant to MUC1. The immune response to the C3-bound MUC1 liposomal vaccine was assessed by ELISA, ELISpot, and flow cytometry. Co-administering TLR 7/8 agonists with MUC1 encapsulated in C3-liposomes resulted in a significant antibody response compared to non-encapsulated MUC1. This antibody response was significantly higher in females than in males. The co-encapsulation of three TLR agonists with MUC1 in C3-liposomes significantly increased antibody responses and eliminated sex-based differences. Furthermore, this immunization strategy resulted in a significantly increased T cell-response compared to other treatment groups. In conclusion, the co-delivery of MUC1 and TLR agonists via C3-liposomes greatly enhances the immune response to MUC1, highlighting its potential for antigen-specific cancer immunotherapy.

## 1. Introduction

Antigen-based cancer vaccines expose antigen-presenting cells (APCs) to immunogenic epitopes from cancer cells, with the goal of stimulating a robust immune response capable of preventing carcinogenesis or reversing tumor growth. Cancer vaccines focus on the delivery of either tumor-specific antigens, proteins with structures that are specific to cancer cells or tumor-associated antigens (TAAs), proteins expressed by cancer cells that are distinct in their expression levels, or post-translational modifications [1,2,3]. Examples of TAAs include the human epidermal growth factor 2, Melanoma-associated antigen 3, and Mucin-1 (MUC1) [4,5]. MUC1 is a highly relevant TAA given its overexpression and hypo-glycosylation in a majority of cancers, and was ranked second among 75 candidate antigens for cancer vaccines by the National Cancer Institute in 2009 [6,7,8].

MUC1 is a transmembrane glycoprotein typically found on the surface of epithelial cells. The C-terminal (intracellular) and N-terminal (extracellular) regions of MUC1 are connected by noncovalent interactions. The extracellular domain includes a variable number of tandem repeats of 20 amino acids (GVTSAPDTRPAPGSTAPPAH) known as the VNTR, which is rich in threonine and serine residues that are potential sites of O-glycosylation. MUC1 is often overexpressed in cancer cells with its carbohydrate residues cleaved from the VNTR regions, enabling the immune system to access the peptide backbone [9,10,11]. These differences in MUC1 expression and glycosylation occur in over 75% of carcinomas, including breast, lung, pancreatic, epithelial ovarian, and prostate cancers, potentially making this protein immunogenic and a promising target for cancer immunotherapy [12]

The effectiveness of a cancer vaccine depends on the recognition of a tumor antigen by T cells and B cells [13]. The activation of these cells is initiated through antigen-presenting cells (APCs), including B cells, macrophages, and dendritic cells, which capture and process antigens for presentation to B and T cells. Given the vital role of APCs, significant efforts have been made to improve the direct delivery of antigens to APCs to enhance the immunogenicity of antigen-based cancer vaccines. One strategy for improving antigen presentation and enhancing immunity against cancer is to encapsulate an antigen within liposomes that target APCs. Some benefits of delivering antigens within liposomes include reduced toxicity, enhanced bioavailability, protection from degradation, and an enhancement in the amount of antigen delivered to APCs [14]. Furthermore, liposomes can also deliver stimulating compounds like toll-like receptor (TLR) agonists to improve the adaptive immune response.

Current nanoparticle antigen delivery techniques to improve the delivery of TLR agonists and tumor antigens include cationic, mannose, Fc-targeted, antibody-targeted and CD11c-targeted liposome carriers [15,16]. While almost all of these liposomal delivery systems have advantages over non-targeted delivery, there are still challenges with the systems. Cationic nanoparticles are attractive because they bind to cell membranes that are negatively charged, but when injected systemically into the body, can aggregate and accumulate in the lung and liver [17]. Many of the actively targeted nanoparticles listed above use antibodies or fragments of antibodies to target dendritic cells. This limits delivery to dendritic cells, and the use of proteins for targeting poses challenges regarding scale up and the storage of the treatment.

In order to target antigen and TLR-based adjuvants to APCs, our laboratory has developed C3-liposomes that utilize endogenous complement C3 to target antigen and TLR agonists directly to C3-receptors on APCs. C3-liposomes contain a lipid with a terminal orthopyridyl disulfide group (OPSS), which readily forms a disulfide bond with the sulfhydryl group of complement C3 upon entering the bloodstream [18,19]. C3-liposomes, coated with complement C3, bind to C3-receptors and enter into cells that express these receptors, allowing for the direct delivery of the antigen and stimulating compounds to APCs. Because OPSS is a small molecule and targeting results from the binding of endogenous C3, many of the storage and scale-up difficulties surrounding antibody targeting are eliminated. Previous studies have shown that C3-liposomes improve antigen presentation and the activation of APCs, resulting in enhanced immune activation and reduced tumor growth [19,20]. TLR agonists including Monophosphoryl-Lipid A (MPLA, specific for TLR4), Resiquimod (R848, specific for TLR7/8), and a single-stranded 20-mer oligodeoxynucleotide (CpG 1826, specific for TLR9) were used for this study based on their similarity to FDA-approved adjuvants. These TLR agonists can increase both the release of immune-stimulating cytokines and the expression of co-stimulatory receptors by APCs [21,22,23].

This study aimed to encapsulate MUC1 peptide and TLR agonists in C3-liposomes and to evaluate the potential of this delivery system to improve the immune response in transgenic mice tolerant to MUC1.

## 2. Materials and Methods

### 2.1. Reagents

A certified clinical-grade 100-amino acid synthetic MUC1 peptide with 5 repeats of the molecular structure of H_2_N-(GVTSAPDTRPAPGSTAPPAH)-CONH_2_ was synthesized at the University of Pittsburgh Peptide Synthesis Facility and was generously provided by Dr. Olivera Finn’s laboratory [24]. The human MUC1 100mer peptide was dissolved in sterile water. Lipids 1,2-dipalmitoyl-sn-glycero-3-phosphocholine (DPPC), 1,2-distearoyl-sn-glycero-3-phosphocholine (DSPC), 1,2-distearoyl-sn-glycero-3-phosphoethanolamine-N-[poly(ethylene glycol)-2000] (DSPE-PEG(2000)), 1,2-distearoyl-sn-glycero-3-phosphoethanolamine-N-[PDP-poly(ethylene glycol)-2000] (DSPE-PEG(2000)-PDP) for liposome production were purchased from Avanti Polar lipids (Alabaster, AL, USA). Lissamine rhodamine B 1,2-dihexadecanoyl-sn-glycero-3-phosphoethanolamine (Rhodamine PE) was purchased from Life Technologies (Grand Island, NY, USA). Complement active mouse serum was obtained from Innovative technologies (Novi, MI, USA). MPLA and R848 were purchased from InvivoGen (San Diego, CA, USA). CpG 1826 was purchased from TriLink (San Diego, CA, USA). CL-4B Sepharose gel for size exclusion chromatography was purchased from Sigma-Aldrich (St Louis, MO, USA). TMB ELISA substrate, Goat anti-mouse IgG H&L, Streptavidin-horseradish peroxidase (HRP), and Mouse anti-human CD227 antibody were purchased from Bio-Rad (Hercules, CA, USA). ChonBlock blocking/sample dilution ELISA buffer and ChonBlock detection antibody dilution buffer were purchased from Chondrex (Woodinville, WA, USA). Flow cytometry antibodies, FITC anti-mouse CD45, PE anti-mouse CD83, PE/DAZZLE anti-mouse CD19, PerCP/Cy55 anti-mouse CD40, PE/Cyanine7 anti-mouse CD11c, APC anti-mouse CD3, Alexa Fluo700 anti-mouse/human CD11b, APC/Cy7 anti-mouse CD8b, BV421 anti-mouse CD86, BV510 anti-mouse Ly-6c, BV605 anti-mouse MHCII, BV650 anti-mouse F4/80, BV785 anti-mouse CD4 were obtained from BioLegend (San Diego, CA, USA). All other reagents were purchased from Thermo Fisher Scientific (Waltham, MA, USA).

### 2.2. Transgenic Mouse Model, Vaccination Groups and Immunization

C57BL/6-Tg(MUC1)79.24Gend/J (MUC1.Tg mice) transgenic mice expressing human MUC1 and C57BL/6J wild-type mice were purchased from the Jackson Laboratory (Bar Harbor, ME, USA). Hemizygous MUC1.Tg males and wild-type females were bred to maintain the colony, with the offspring genotyped using PCR to identify hemizygous male and female MUC1.Tg mice. All mice used in the experiments were housed in the animal facility at the University of Alaska, Anchorage. Four groups of mice were used for this study, with 4 male and 4 female MUC1. Tg mice (8–12 weeks old) in each group. Mice were vaccinated with the following vaccine formulations: Group 1: MUC1 encapsulated in C3-liposomes and non-encapsulated R848 (MUC1 C3-liposomes/R848); Group 2: Non-encapsulated MUC1 and non-encapsulated R848 (Free MUC1/R848); Group 3: MUC1 with the triple TLR agonists MPLA, R848, and CpG 1826 encapsulated in C3-liposomes (MUC1 C3-liposomes/3Adj); Group 4: Non-encapsulated MUC1 with the triple TLR agonists MPLA, R848, and CpG 1826 encapsulated in non-targeted control liposomes (Free MUC1+control liposomes/3Adj). Vaccines were injected subcutaneously in the left flank on day one, followed by a booster shot on day seven. On day fourteen, the mice were euthanized. Blood and spleen were collected and analyzed by flow cytometry, ELISA, and ELISpot. All experiments were approved by the UAA Institutional Animal Use and Care Committee, and mice were monitored daily for any signs of distress.

### 2.3. Liposome Preparation

Liposomes were prepared by film hydration–extrusion, as previously described [18,19]. Lipids DPPC/DSPC/DSPE-PEG (2000)-PDP/DSPE-PEG (2000) with the ratio of 76:18:3:3 were mixed to produce C3-liposomes. For the control liposomes, DSPE-PEG (2000)-PDP was left out of the formulation, and lipids were mixed at the ratio of 76:18:6. Briefly, to produce liposomes, lipids were dissolved in chloroform, dried under nitrogen for one hour, rehydrated with 0.7 mL of DI water containing MUC1 (1 mg/mL), and extruded nine times at 47 °C through an Avanti Mini Extruder (Avanti Polar Lipids). For liposome formulations containing TLR agonists, 60 μL of MPLA (1 mg/mL) was added to the lipids before drying under nitrogen; then, a mixture of CpG 1826 (4.5 mg), R848 (1 mg), and 100 mer MUC1 (1 mg/mL) in a total volume of 0.7 mL DI water was added for rehydration. For liposome internalization experiments, lipid formulations contained 1% Rhodamine PE as a fluorescent tag. Liposomes were separated from non-encapsulated material using a CL-4B Sepharose column hydrated in PBS, pH 7.4, and the liposome size was determined with a Malvern Zetasizer Nano-S (Malvern Instruments, Malvern, UK). The antigen encapsulation efficiency was determined using HPLC analysis, as described below.

### 2.4. HPLC-DAD Analysis of MUC1

MUC1 C3-liposomes were analyzed using a 1200 Infinity series liquid chromatography system with diode array detection controlled by MassHunter v. B.06.00 (Agilent, Santa Clara, CA, USA). First, 10 µL injections were made using a Zorbax 300SB-C8 column (100 × 2.1 mm, 3.5 µm) (Agilent, Santa Clara, CA, USA) maintained at 35 °C. MUC1 was eluted with a gradient of (A) water and (B) acetonitrile acidified with 0.1% trifluoroacetic acid at a flow rate of 0.3 mL/min. The mobile phase composition was increased from 10 to 95% acetonitrile over 10 min and held for 3 min before re-equilibrating to initial conditions. The retention time and total run time were 5.71 min and 20 min, respectively. MUC1 was detected via absorbance at 220 nm with a reference wavelength of 240 nm; the bandwidths were 4 and 20 nm, respectively. The amount of MUC1 encapsulated in the liposomes was determined to be 10–15 μg/mL for all formulations.

### 2.5. Targeting and Internalization of Liposomes

To evaluate the uptake of C3-liposomes by CD11b^+^ immune cells, bone marrow cells were extracted from naïve wild-type C57BL/6J mice, incubated with fluorescently tagged liposome formulations, and analyzed by fluorescence microscopy and flow cytometry. To conjugate complement C3 to the OPSS group, 10 μL of liposomes was incubated with 10 μL of complement active mouse serum at room temperature for 1 h. Bone marrow cells (6 × 10^5^ cells) in 80 μL of RPMI medium without serum were added to the mixture, bringing the final volume to 100 μL (10% serum), which was plated in a 96-well clear-bottom plate (Becton Dickinson Labware, Franklin Lakes, NJ, USA). The cells were incubated at 37 °C and 5% CO_2_ for 3 h, adherent cells were rinsed with 100 µL of PBS, and cells were observed with an inverted fluorescence microscope, Leica DMI6000B (Leica Microsystems, Buffalo Grove, IL, USA). Photographs were taken using a 20x objective utilizing Leica Application Suite version 3.7.0 software (Leica Microsystems, Wetzlar, Germany). The cells prepared as described here were also analyzed by flow cytometry, using previously described techniques [19]. Cells were stained with fluorescent antibodies to mouse CD45, CD3, CD19, CD11b, CD4, CD8 (BioLegend) and analyzed with a Beckman Coulter CytoFLEX flow cytometer with CytExpert software, version 2.0.0.153 (Beckman Coulter, Brea, CA, USA).

### 2.6. ELISA

Serum samples from vaccinated mice were analyzed for MUC1-specific IgG antibody responses using ELISA assays. A 96-well flat-bottom plate (Invitrogen^TM^ Nunc MaxiSorp, Thermo Fisher Scientific (Waltham, MA, USA) was coated with 100 μL of 1 μg/mL 100-mer MUC1 in PBS/well and incubated at 4 °C overnight. Serum samples from vaccinated mice were added at an appropriate dilution (1/100). The ELISA was run using ChonBlock solutions according to the supplied manufacturer protocol. Mouse anti-human CD227 antibodies were used to generate a standard curve, and non-coated wells were used as negative controls. The absorbance was read at 650 nm using a Molecular Devices Spectramax iD3 ELISA plate reader (Molecular Devices, San Jose, CA, USA).

### 2.7. ELISpot

A murine interferon-gamma (IFN-γ) single-color enzymatic ELISpot kit (ImmunoSpot kit, Cleveland, OH, USA) was used to quantitate IFN-γ-producing T cells against the MUC1 antigen in vaccinated mice. Briefly, mouse spleens were digested with collagenase (1 mg/mL), and the number of spleen cells were counted. A volume of 6 μL of 100 mer MUC1 (1 mg/mL) in 94 μL of CTL-Test^TM^ medium was mixed with 100 μL of spleen cells (800 × 10^5^ cells) and seeded into each well. The plate was incubated at 37 °C and 9% CO_2_ for 24 h. The next day, antibodies were used to detect the number of T cell clones producing IFN-γ, according to the ImmunoSpot Elispot kit protocol. After overnight drying at room temperature, the plate was read and analyzed using a CTL ImunoSpot S6 Micro analyzer (ImmunoSpot, Cleveland, OH, USA).

### 2.8. Flow Cytometry Analyses on Spleen Cells

After the collagenase digestion of spleens, immune cells were analyzed by flow cytometry as previously described [19]. FACS buffer-suspended spleen cells were stained with fluorescent antibodies to murine CD45, CD25, CD19, CD40, CD11c, CD3, CD11b, CD8, CD86, Ly6c, MHC II, F4/80, and CD4, and analyzed with a Beckman Coulter CytoFLEX flow cytometer using the CytExpert software, version 2.0.0.153 (Beckman Coulter, Brea, CA, USA).

### 2.9. Statistical Analysis

Statistical analysis was performed using GraphPad Prism software (Version 9.5.0) and the R studio program. Student’s unpaired *t*-test was performed to compare the means of two independent or unrelated groups. All analyses were considered statistically significant when the *p* value was 0.05 or less.

## 3. Results

### 3.1. MUC1 C3-Liposomes Are Internalized by CD11b^+^ Immune Cells

Our prior research has shown that the OPSS group in C3-liposomes facilitates the binding of complement C3 and results in the internalization of liposomes into APCs via C3 receptors [19]. To verify that the encapsulation of three TLR agonists (R848, CpG, and MPLA) into liposomes does not interfere with targeting, the internalizations of MUC1 C3-liposomes/3Adj and MUC1 control liposomes/3Adj were compared with liposome formulations that did not have the three TLR agonists.

There was no significant difference in liposome size for the various formulations. MUC1 C3-liposomes/3Adj had a diameter of 279 ± 138 nm and a polydispersity index (PDI) of 0.160. Control liposomes/3Adj had a diameter of 258 ± 59 nm with a PDI of 0.239, and MUC1 C3-liposomes without adjuvant had a diameter of 266 ± 94 nm with a PDI of 0.108. MUC1 was trapped in liposomes through passive encapsulation, with liposomes containing 10−15 μg/mL of MUC1 for all formulations, representing an encapsulation efficiency of 3.3–5.0%.

Fluorescence microscopy images revealed that both rhodamine-labeled MUC1 C3-liposomes and MUC1 C3-liposomes/3Adj were taken up by bone marrow cells, while those groups that did not contain OPSS-conjugated lipids were not engulfed (Figure 1A). The analysis of cells by flow cytometry showed that MUC1 C3-liposomes were readily taken up by CD11b^+^ cells (60.88 ± 1.23), while there was little uptake of the MUC1 control liposomes (5.053 ± 7.00) (Figure 1B–D). Likewise, MUC1 C3-liposomes/3Adj showed a higher uptake into CD11b^+^ cells (85.59 ± 2.26) than MUC1 control liposomes/3Adj (3.59 ± 2.1), demonstrating that the inclusion of TLR agonists in the C3-liposomal formulation still allows for the targeting of myeloid APC precursors (Figure 1C,D). It is interesting to note that CD11b^+^ cells internalized MUC1 C3-liposomes/3Adj significantly more than MUC1 C3-liposomes (Figure 1D).

### 3.2. MUC1 C3-Liposomes with Encapsulated TLR Agonists Induce a MUC1-Specific IgG Antibody Response in MUC1 Transgenic Mice

To determine whether MUC1 C3-liposomes enhanced antibody responses in vivo, female and male MUC1.Tg mice were vaccinated with non-encapsulated R848 and either MUC1 C3-liposomes or Free MUC1 at the same concentration. Mice treated with MUC1 C3-liposomes/R848 generated significantly more anti-MUC1 IgG antibodies than those treated with Free MUC1/R848 (193 vs. 76 μg/mL) (*p* value = 0.03) (Figure 2a). To assess the in vivo effects of multiple TLR agonists encapsulated in liposomes, three TLR adjuvants (R8484, CpG, and MPLA) were encapsulated into a C3-targeting liposomal formulation (MUC1 C3-liposomes/3Adj) versus a non-targeted liposomal formulation (Free MUC1+ control liposomes/3Adj), and the liposomes were injected into MUC1.Tg mice. When TLR agonists were included in the liposomal formulation, MUC1 C3-liposomes/3Adj induced the production of significantly more anti-MUC1 IgG antibodies compared to MUC1 C3-liposomes/R848 (315.1 vs. 193.3 μg/mL) (*p* value = 0.0410) and compared to Free MUC1/R848 (315.1 vs. 72.96 μg/mL) (*p* value < 0.0001) (Figure 2a). There was no significant difference between the antibody titers of the mice treated with targeted MUC1 C3-liposomes/3Adj versus those treated with Free MUC1+control liposomes/3Adj, demonstrating that encapsulated TLR agonists do not necessarily need to be targeted to increase the antibody production (315.1 vs. 283.4 μg/mL) (*p* value = 0.5342) (Figure 2a).

### 3.3. Delivery of MUC1 with TLR Agonists in C3-liposomes Eliminated Sex Differences in the Anti-MUC1 Antibody Response

Interestingly, the IgG antibody response following vaccination showed sex-based differences in the antibody titer. Female mice vaccinated with MUC1 C3-liposomes/R848 generated significantly higher anti-MUC1 IgG antibody responses compared to males (305.3 vs. 81.32 μg/mL) (*p* value = 0.0031) (Figure 2b). However, when TLR agonists were encapsulated within C3-liposomes, the sex-based difference was reduced due to a significant increase in the anti-MUC1 IgG antibody titer in male mice (Figure 2b). Male mice that received MUC1 C3-liposomes/3Adj produced significantly higher amounts of anti-MUC1 IgG antibodies compared to male mice that received MUC1 C3-liposomes/R8484 (270.9 vs. 81.32 μg-/mL) (*p* value = 0.0009). In contrast, no significant differences in antibody response were observed in female mice vaccinated with MUC1 C3-liposomes/R848 versus MUC1 C3-liposomes/3Adj (Figure 2b).

### 3.4. Vaccination with MUC1 C3-Liposomes and TLR Agonists Induces a Robust Antigen-Specific T Cell Response

To evaluate the T cell response after vaccination, spleen cells from vaccinated MUC1.Tg mice were isolated, stimulated with MUC1 antigen, and evaluated for IFN-γ production using ELISpot assays. Mice vaccinated with MUC1 C3-liposomes/3Adj produced significantly more IFN-γ than mice treated with MUC1 C3-liposomes/R848, free MUC1/R848, or free MUC1+Control-liposomes/3Adj (Figure 3a). Based on these results, the encapsulation of TLR agonists in C3-liposomes leads to the increased activation of IFN-γ-producing MUC1 antigen-specific T cells (Figure 3a). T cell production of IFN-γ did not appear to be sex-based, with males and females showing significantly greater T cell responses after vaccination with MUC1 C3-liposomes/3Adj (Figure 3b).

### 3.5. Analysis of Splenic Lymphocyte Populations in Vaccinated Mice

Flow cytometry was used to determine the impact of MUC1 C3-liposomes and TLR agonists on the overall B cell, T cell and myeloid cell populations in the spleens of vaccinated mice. The percentages of T (CD3^+^) and B (CD19^+^) lymphocytes and of myeloid cells (CD11b^+^) in the spleen were similar in mice vaccinated with MUC1 C3-liposomes/3Adj versus those vaccinated with Free MUC1 + control liposomes/3Adj (Figure 4). While the number of IFN-γ-secreting T cells against MUC1 increased in the ELISpot analysis (Figure 3), the percentages of T cells, B cells and myeloid cells in the spleen were the same between groups (Figure 4). Breaking down the T cells into CD4^+^ and CD8^+^ phenotypes and the myeloid cells into antigen-presenting (MHCII^+^) and immature (MHCII^-^Ly6C^+^) phenotypes showed that there were also no significant differences in the treatment groups regarding these cell types (Figure 4).

## 4. Discussion

MUC1 vaccine formulations expose APCs to MUC1 with the goal of boosting antibody and T cell responses to cancer cells. Successful examples of MUC1 vaccines include MUC1-loaded dendritic cells, MUC1 peptides, and liposomal MUC1 peptide-based vaccines [25,26,27]. Strategies based on dendritic cells loaded with MUC1 have effectively stimulated MUC1 antigen-specific CD8+ T cells that secrete IFN-γ but do not seem to produce detectable antibodies [28,29]. Finn et al. developed several vaccines based on MUC1 peptides using a 100mer peptide, which induced both a T cell and antibody response [24,25,30]. Mice that received the MUC1 vaccine had a lower risk of developing colitis-associated colon cancer due to a significant increase in anti-MUC1 IgG antibodies and MUC1 antigen-specific T cells [31].

Our research aimed to utilize the MUC1 100mer developed in Dr. Olivera Finn’s lab to enhance B and T cell responses using C3-liposome-targeted delivery to APCs. Because MUC1 is a self-antigen with low immunogenicity compared to neoantigens, several groups have explored the possibility of boosting vaccination by adding TLR agonists [32,33]. We hypothesized that the encapsulation of MUC1 and multiple TLR agonists into C3-liposomes would enhance humoral- and cell-mediated responses in vaccinated mice due to the improved targeting and activation of APCs.

Fluorescently tagged C3-liposome formulations with encapsulated TLR agonists were internalized by CD11b^+^ bone marrow cells, as shown by fluorescence microscopy and flow cytometry. The results demonstrated that rhodamine-labeled MUC1 C3-liposomes and MUC1 C3-liposomes/3Adj were taken up by CD11b^+^ cells, while control liposomes without OPSS-conjugated lipids showed very little internalization. These findings correlate with our previous studies and confirm the role of C3-complement binding to OPSS-conjugated lipids in enhancing the delivery of MUC1 peptides by C3-liposomes [19]. Interestingly, the internalization of MUC1 C3-liposomes/3Adj was significantly higher than that of MUC1 C3-liposomes. It is possible that MPLA, which is included in this liposome formulation, binds to its TLR4 receptor on CD11b^+^ cells and augments the C3-driven binding of liposomes, resulting in a higher internalization rate of MUC1 C3-liposomes/3Adj [30].

In MUC1.Tg mice, the MUC1 C3-liposomes/R848 vaccine formulation with encapsulated MUC1 induced significantly higher antibody responses than the Free MUC1/R848 vaccine. Interestingly, female mice vaccinated with MUC1 C3-liposomes/R848 produced significantly higher amounts of anti-MUC1 IgG antibodies compared to male mice vaccinated with the same formulation, which aligns with results found in other vaccine studies [34,35,36]. Multiple mechanisms, such as hormonal and genetic, may contribute to sex differences in vaccine-induced immunity [37,38]. In females, heightened levels of estrogen have been correlated with increasing levels of antibodies, such as during pregnancy, suggesting that estrogen plays a role in antibody production [39]. In males, high testosterone levels were correlated with lower antibody responses to influenza vaccines [40]. Furthermore, genetic and epigenetic differences between females and males could also influence humoral immunity. Genes involved in regulating pathogen and vaccine-specific immunity are located on the X-chromosome (such as TLR7 signaling), or genes can be modulated in a sex-dependent manner due to estrogen response elements (EREs) in their promotors, supporting enhanced immune responses to viral infections and vaccines in females [39,41]. Based on our results and those of other groups, further research on vaccines using C3-liposomes should continue to look at sex-based differences and analyze both antibody titers and the ability of antibodies to bind to their target in male and female mice.

Our results showed that the encapsulation of three TLR agonists within MUC1 C3-liposomes eliminated the sex-based differences in the antibody titer by enhancing the antibody response in male mice but not in female mice. These results suggest that the TLR-mediated co-stimulation of APCs can enhance IgG antibody production by B cells [42]. TLR7 and TLR9 are co-expressed on B cells, and it is likely that introducing TLR agonists to these receptors in liposomal formulations helped drive B cell activation and antibody production [43]. The addition of a TLR4 ligand to the vaccine may also have helped boost antibody production, as TLR4 plays a vital role in promoting antibody production by upregulating the expression of B-cell-activating factor (BAFF) [33,44]. The strong ability of TLR agonists to boost the antibody response in male mice was apparent, given that vaccine formulations containing TLR agonists stimulated similarly high antibody responses regardless of the delivery strategy.

T cells play a crucial role in cancer immunity. A major goal in the development of cancer vaccines is to induce a robust tumor antigen-specific T cell response, particularly in cytotoxic CD8+ T cells, to prevent the development of cancer. Our prior studies have shown that antigens encapsulated within C3-liposomes can activate a T cell response in mouse models of cancer [20]. Our current study tested MUC1-containing vaccine formulations for their ability to induce MUC1 antigen-specific IFN-γ-producing T cells. Our results indicate that the encapsulation of multiple TLR agonists in MUC1 C3-liposomes boosted IFN-γ-producing T cells in both vaccinated male and female mice. Compared with the control groups, it appears that targeting both the MUC1 antigen and TLR agonists is required for an enhanced T cell response. The targeted delivery of the MUC1 100mer peptide and TLR agonists to APCs by our C3-liposomes likely resulted in a specific costimulatory activation of APCs, which in turn resulted in the activation of MUC1 antigen-specific IFN-γ-producing T cells.

## 5. Conclusions

The data reported here provide evidence that the encapsulation of the MUC1 100mer peptide and multiple TLR agonists in C3-liposomes elicits robust antigen-specific cellular and humoral immune responses in MUC1 transgenic mice. Given this immune response, a C3-liposome-based vaccine formulation could possibly be used prophylactically as a vaccine to prevent the development of cancers with aberrant MUC1 expression, as well as in cancer treatment settings to enhance the immune recognition of MUC1-expressing cancer cells.

## Figures and Tables

**Figure 1 pharmaceutics-15-02774-f001:**
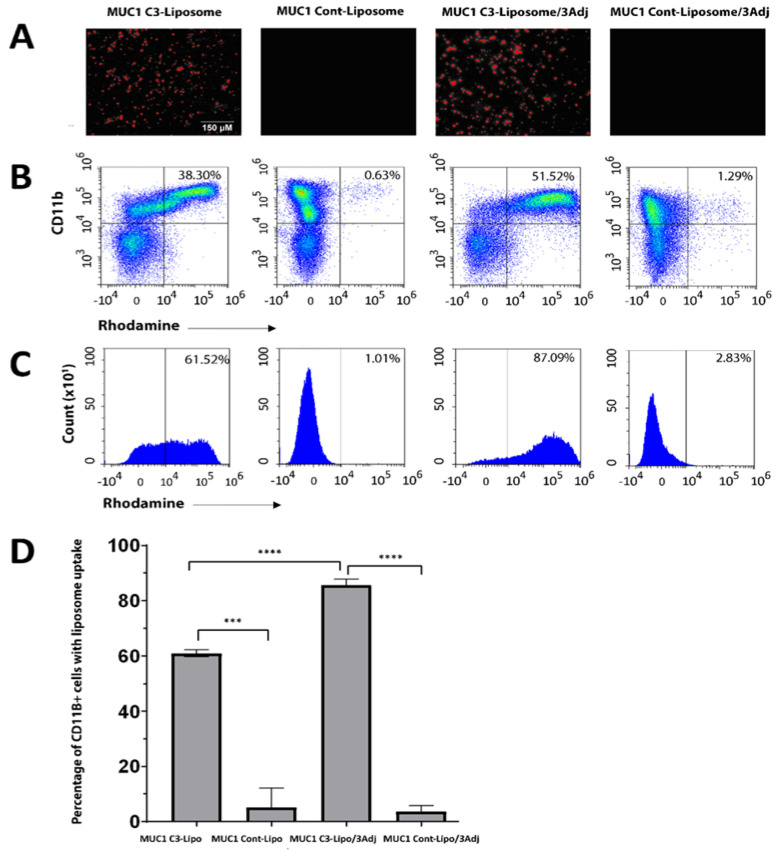
CD11b^+^ immune cells internalize MUC1 C3-liposomes. (**A**) Representative fluorescence microscopy images of cellular uptake of rhodamine+ labeled liposomes revealed that MUC1 C3-liposomes and MUC1 C3-liposomes/3Adj have a higher uptake than MUC1 control liposomes and MUC1 control liposomes/3Adj by murine bone marrow cells. (**B**) Representative flow cytometry dot plots show percentages of double-positive CD11b^+^ and rhodamine+ cell subsets for each group. (**C**) Representative histogram shows the percentage of CD11b^+^ cells that have taken up rhodamine-labeled liposomes for each group. (**D**) Quantification of the percentage of CD11b^+^ rhodamine+ cells by flow cytometry revealed a significantly higher uptake of MUC1 C3-liposomes and MUC1 C3-liposomes/3Adj by CD11b^+^ cells compared to MUC1 control liposomes (*** *p* value = 0.0002) and MUC1 control liposomes/3Adj (**** *p* value < 0.0001). Data are presented as Standard Error of the Mean (n = 3). 3 Adjuvants (3Adj) = MPLA, CpG and R848.

**Figure 2 pharmaceutics-15-02774-f002:**
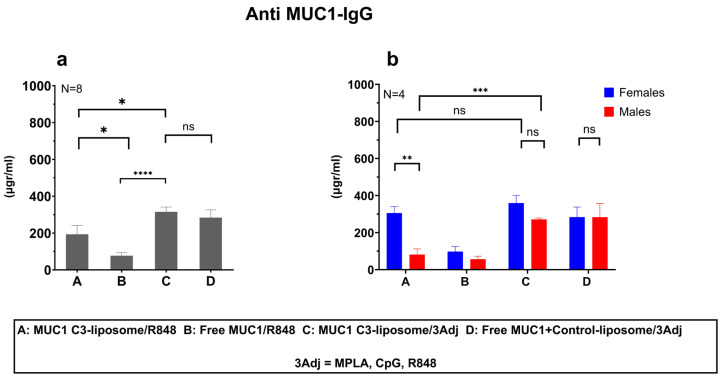
MUC1 encapsulated with TLR agonists in C3-liposomes induced an anti-MUC1 IgG antibody response in mice. (**a**) MUC1 C3-liposomes/R848 and MUC1 C3-liposomes/3Adj elicited higher antibody responses compared to Free MUC1/R848 in MUC1.Tg mice. (**b**) There was a significant difference between females and males in the MUC1 C3-liposomes/R848 group, but the encapsulation of three TLR agonists eliminated this sex-based difference. Data are presented as Standard Error of the Mean. *p*-value: ns > 0.05, * ≤ 0.05, ** ≤ 0.01, *** ≤ 0.001, **** ≤ 0.0001. N = 8 (4 male and 4 female mice per group) < 0.05. N = 8 (4 male and 4 female mice per group).

**Figure 3 pharmaceutics-15-02774-f003:**
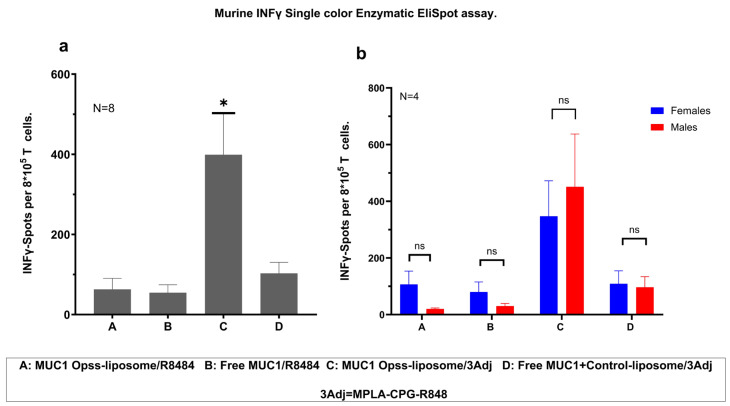
Encapsulation of MUC1 and triple adjuvants in C3-liposomes increases T cell response. (**a**) ELISpot analysis showed a significantly higher level of IFN-γ-secreting T cells in MUC1.Tg mice that received MUC1 C3-liposomes/3Adj compared to other groups (N = 8 mice per group). (**b**) There were no significant differences between males and females in any of the groups. Data are presented as Standard Error of the Mean. * *p*-value < 0.05 indicates a significant difference between the groups, N = 8 (4 females and 4 males per group).

**Figure 4 pharmaceutics-15-02774-f004:**
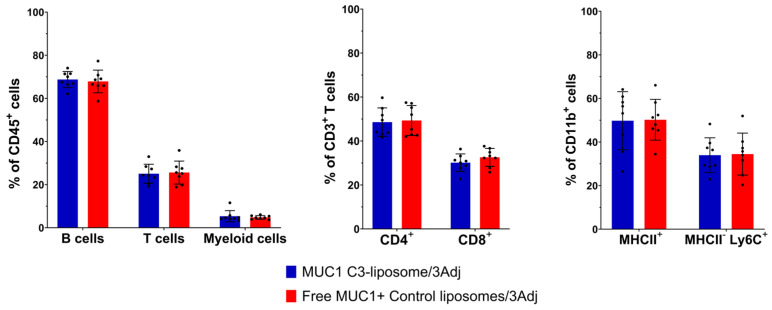
Flow cytometry analysis of B cells, T cells and myeloid cells in the spleen of vaccinated mice. The percentages of B (CD19^+^), T cells (CD3^+^) and myeloid cells (CD11b^+^) are shown for the spleen white blood cells of mice vaccinated with encapsulated MUC1 C3-liposomes/3Adj versus Free MUC1+Control-liposomes/3Adj. The T cells are further broken into CD4^+^ and CD8^+^ phenotypes, and the myeloid cells are broken down into an antigen-presenting phenotype, MHCII^+^, and a suppressive phenotype, MHCII^-^Ly6C^+^. Mean ± standard error, N = 8 (4 males and 4 females). 3Adj = MPLA, CpG and R848.

## Data Availability

The data presented in this study are available in this article.

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
