# Peer review of "Enhancing T Cell and Antibody Response in Mucin-1 Transgenic Mice through Co-Delivery of Tumor-Associated Mucin-1 Antigen and TLR Agonists in C3-Liposomes"

_pharmaceutics, 2023, doi:10.3390/pharmaceutics15122774_

Round 1

Reviewer 1 Report

Comments and Suggestions for Authors

This manuscript investigated co-delivery of MUC1 antigen and adjuvants using C3-binding liposomes towards induction of antigen-specific immune responses. Three TLR adjuvants and MUC1 antigen were incorporated into the liposomes having binding capability to C3. Conjugation of C3 onto the liposomes significantly improved the cellular association of liposomes into monocytes. After immunization of these liposomes, MUC1-specific antibody responses and T cell responses in the spleen were induced. Furthermore, sex-dependence of immune responses was also observed. Therefore, this manuscript would provide useful information on the immunity induction approach for non-immunogenic antigen by using functional liposomes. However, current manuscript still requires some corrections as listed below:

-Introduction: Before introduction of the purpose of this study (line 86-), relating researches using functional liposomes containing various adjuvants or other targeting approaches using liposomes to further increase the rationality and superiority of C3-liposomes compared with relating functional liposome-based antigen delivery systems.

-Results: There is no characterization information for prepared liposomes including size, PDI, zeta-potential, and encapsulation efficacy of MUC1 per liposomes.

-Figure 4: The comparison of B cell and T cell populations between PBS-treated group and liposome-treated groups would be helpful to understand immunity induction mechanism of liposomes. Furthermore, analysis on dendritic cell population as a representative antigen presenting cells and the populations of T cell subsets (effector T cell, central memory cell and so on) would provide more precise data to analyze immune responses induced by the liposomes.

Reviewer 2 Report

Comments and Suggestions for Authors

In this manuscript, the author described a C3-liposome strategy to co-encapsulate MUC1 antigen and TLR agonists. The MUC1 C3-liposomes can greatly enhances the immune response to MUC1, highlighting its potential for antigen-specific cancer immunotherapy. The manuscript predominantly focuses on immune-related content but lacks depth in pharmaceutical studies. Consider expanding on pharmaceutical aspects, including specific details related to liposome characteristics and encapsulation efficiency. The following are our specific comments for this manuscript:

1.     Acronym Usage:

It is recommended to avoid acronyms or abbreviations if they are used only once. Examples include:

Line 22: TLR

Line 52: VNTR

Line 132: CpG

Line 211: IFN-γ

2.     Typographical Correction:

In line 186, please correct "CO2" to "CO2."

3.     Figure 1 Clarification:

In Figure 1, the term "Adj" in the "MUC1 C3-Lipo / 3 Adj" group needs clarification. If it refers to adjuvants, please explicitly state so in the manuscript for reader understanding.

4.     Particle Size Measurement:

Mention that the manuscript discusses the use of the Malvern Zetasizer Nano-S to measure liposome particle size, but specific data on the particle size is not provided. Please include this data for a comprehensive presentation.

5.     Statistical Variation in Figure 3:

Consider addressing the relatively large standard deviation (SD) values in Figure 3 by potentially increasing the number of mice in each group to improve statistical robustness.

6.     Liposome Concentration and Encapsulation Efficiency:

The article states the liposome concentration as 10-15 μg/mL but lacks specific data on encapsulation efficiency. Including this information is crucial for evaluating liposome quality and providing practical references for preparation applications (line 174).

7.     Clarity in HPLC Elution Conditions:

Provide more specific and detailed information about the elution conditions of HPLC at the end of the article for enhanced clarity.

8.     Antigen Encapsulation Efficiency:

The manuscript does not mention antigen encapsulation efficiency, and the origin of the MUC1 amount in liposomes is unclear. Please provide relevant data to address these concerns.

9.     Pharmaceutical Investigation:

The article lacks essential pharmaceutical investigations of C3 liposomes, such as particle size distribution, electron microscope images, and other relevant details. Including this information would strengthen the pharmaceutical aspects of the study.

Comments on the Quality of English Language

I do not have specific comments on the English writings.

Reviewer 3 Report

Comments and Suggestions for Authors

The team of authors has carried out an important and very interesting study in the development of liposomal cancer vaccines based on the induction/enhancement of the immune response to mucin-1. I believe that this study, although not revolutionary or fundamental, but does contain a number of interesting conclusions and observations that will be of interest to researchers in the field of cancer vaccine development and related fields. Meanwhile, authors should answer the following questions:

1. What adjuvant was used to enhance the effect of the vaccine?

2. According to the authors, what is the mechanism of increased antibody synthesis in males and low antibodygenesis in female mice?

3. Has interferon status been studied in males and females?

4. What line of mice was used? Be sure to provide a complete description of the mouse line in Materials and Methods.

Round 2

Reviewer 1 Report

Comments and Suggestions for Authors

The authors revised the manuscript properly according to the reviewer's comments.